# Machine learning assisted differentiation of low acuity patients at dispatch: The MADLAD randomized controlled trial

Douglas Nils Spangler[1]*, Simon Morelli[2], David Smekal[1], Lennart Edmark[1,2], Hans Blomberg[1]

1 Department of Surgical Sciences, Uppsala University, Uppsala, Sweden, 2 Centre for Clinical Research, Västmanland Hospital, Västerås, Sweden

* douglas.spangler@akademiska.se

## Abstract

### Background

Resource Constrained Situations (RCS) at Emergency Medical Dispatch centers where there are more patients requiring an ambulance than there are available ambulances are common. Machine Learning (ML) techniques offer a promising but largely untested approach to assessing relative risks among these patients. The study aims to establish whether the provision of ML-based risk scores predicting patient outcomes improves the ability of dispatchers to identify patients at high risk for deterioration in RCS and dispatch the first available ambulance to them.

### Methods and findings

We performed a parallel-group, randomized trial of adult patients assessed by a dispatch nurse at two study sites in Sweden as requiring a low-priority ambulance response in RCS. Patients were randomized 1:1 to be prioritized with the aid of an ML-based risk assessment tool, or per current clinical practice. The primary outcome was defined in terms of whether the first available ambulance was sent to the patient with the highest National Early Warning Score (NEWS 2) based on subsequently collected vital signs. A total of 1,245 RCS were included in the study. In the intervention arm, 68.3% of RCS were assessed correctly per the primary outcome versus 62.5% in the control group, corresponding to an odds ratio of 1.28 (95% CI [1.00, 1.63], $p = 0.047$). This study was limited to only patients determined to require a low-priority ambulance response in two Swedish regions, and was underpowered for the primary outcome due to a smaller than expected sample size.

### Conclusion

This study suggests that clinical ML-based decision support tools may have the ability to influence care provider decisions and improve their capacity to rapidly

**Data availability statement:** The data used in this study are owned by the regional health authorities in each respective study site, and permission to publicly distribute individual level data was not granted by the Swedish ethics review authority. The data used in this study may be obtained by researchers with appropriate ethics approvals by contacting ambulanss-jukvard@akademiska.se. All code used to generate the reported results is available in a public repository at https://doi.org/10.17605/OSF.IO/ERKV7 and the tool evaluated is available at https://github.com/dnspangler/opentriage.

**Funding:** Partly funded by a grant to HB from the Swedish Innovation Agency (vinnova.se) grant number 2017-04652. The funders had no role in study design, data collection and analysis, decision to publish, or preparation of the manuscript.

**Competing interests:** The authors have declared that no competing interests exist.

differentiate between high- and low-risk patients at dispatch. Further research should establish the suitability of these tools in larger cohorts, for patients with both higher- and lower-levels of priority, and in other settings. The trial was registered at Clinical-Trials.gov (NCT04757194).

---

## Author summary

### Why was this study done?

- The substantial time pressure and uncertainty faced by emergency medical dispatchers entails a substantial need for improved decision support tools.

- There are few randomized trials of machine learning based decision support tools in emergency medicine, which are necessary to robustly establish the efficacy of such tools.

### What did the researchers do and find?

- A machine learning (ML) based tool was investigated in a randomized controlled trial, evaluating its ability to improve prioritization decisions in resource constrained situations where the number of patients awaiting an ambulance exceeded the number of available ambulances.

- 1,245 such situations were included in the trial, and the study found that the intervention was associated with an increased proportion of cases where the first available ambulance was dispatched to the most critically ill patient from 62.8% to 68.3%.

### What do these findings mean?

- The study shows that ML based risk assessment tools may improve the ability of dispatchers to rapidly and accurately prioritize patients in the context of emergency medical dispatching.

- The study was limited to adult patients determined to require a low-priority ambulance response, and further research is needed to establish efficacy in both high-priority patients, and those determined not require an ambulance.

- Given the limited sample size and restricted setting of two Swedish regions, these findings should be validated larger in cohorts and more diverse settings.

## Introduction

In prehospital care systems, ambulance availability places constraints on the ability of Emergency Medical Dispatch (EMD) centers to immediately provide an ambulance response for all patients determined to require one. The stochastic nature of

ambulance demand via the emergency hotline entails that any cost-effective ambulance system will from time to time experience Resource Constrained Situations (RCS) in which the number of patients requiring an ambulance response exceeds the number of available ambulances (i.e., where two or more patients are simultaneously waiting for an ambulance to become available). In such cases, one patient must be prioritized to receive an immediate ambulance response (i.e., to have the first available ambulance dispatched to them), while the remainder will receive a further delayed response.

Machine Learning (ML) models offer a promising approach to stratifying the risks associated with these patients. In previous research by the authors, an open-source ML-based risk assessment tool was developed and retrospectively validated in a cohort of patients receiving ambulance care in the region of Uppsala [1,2]. A number of ML-based risk assessment tools have been proposed for use in prehospital and emergency care and validated, retrospectively [3–9]. Only a single randomized trial has been identified in recent systematic reviews [10,11], and no models intended for risk differentiation across the full spectrum of prehospital patient types were found to have been evaluated in a randomized trial. There is thus a great need to identify suitable use-cases for these tools, and to generate high-quality evidence regarding their effectiveness in achieving clinically important objectives. Given that the application of ML models in this context is novel and unproven, this must be done in a manner which minimizes the patient safety hazards associated with incorrect decisions.

This study aims to investigate whether the application of a ML-based risk assessment instrument improves the ability of dispatchers to identify and dispatch an ambulance to the most acute patient in an RCS.

## Methods

### Design

A parallel-group trial, randomized 1:1 to intervention or control arms.

### Setting

The study took place in two EMD centers in central Sweden (Uppsala and Västmanland), serving a combined population of 499,000 in 2021. The regions have a total of 32 ambulances during peak hours. Each dispatch center is staffed by 2–3 dispatch nurses that answer emergency (112) and nonemergency calls, and 1 ambulance dispatcher 24 hours per day. The dispatch nurses currently employ a self-developed, rule-based Clinical Decision Support System (CDSS) labeled *Medicinska Beslutsstödet* (MBS) to structure patient interviews and determine a priority level [12,13]. The CDSS results in a priority of 1A/1B (lights and sirens response), 2A/2B (nonemergency response), or a recommendation of referral to nonambulance care. Priority 2A/2B patients were chosen as the target population for the intervention, given that they had been determined by a dispatch nurse to have relatively low acuity, but would also not be exposed to the risks associated with being referred to nonemergency care [12,14,15]. In the studied setting, RCS also predominantly occur in these priority 2 cases, as an ambulance is typically reserved to immediately respond to any incoming calls determined to meet the criteria for a priority 1 response.

### Participants

Participants were recruited between 2021-02-01 and 2024-12-01, with data collection at the second site beginning in 2024-06-01. A data monitoring committee was formed to monitor the study and address any reported incidents or patient safety issues, but none were identified. The study was ended by the authors prior to having collected the full calculated sample size (1,500) for administrative reasons due to the need to include the study in a doctoral thesis. The criteria for inclusion were:

• Identification of a RCS by ambulance dispatcher.

• Assigned priority 2A or 2B by dispatch nurse.

- Complete call documentation in the CDSS.

- Valid Swedish personal identification number collected at dispatch.

- Age ≥ 18 years.

In addition to these inclusion criteria, dispatchers screened for exclusion criteria which would render a direct comparison of clinical risks less relevant:

- Relevant calls received more than 30 min apart.

- Logistical factors (e.g., the patients' geographical locations) affect the ambulance assignment decision.

- On-scene risk factors (e.g., a patient is outdoors and risks hypothermia) or risk mitigators (e.g., healthcare staff already on-scene with a patient) affect the ambulance assignment decision.

Following the application of these criteria by dispatchers, post-randomization exclusions were applied to remove patients who retroactively declined to participate in the study, and randomizations where no ambulance was dispatched prior to the inclusion of the patients in another randomization, resulting in a participant flow per Fig 1 below.

### Intervention

Ambulance dispatchers had overall responsibility for executing the study protocol, and were tasked with identifying RCSs suitable for inclusion in the study. Patients were to be included in the study upon the identification of an RCS involving eligible patients by the ambulance dispatcher at the point in time when an ambulance was available for dispatch to one of the patients. Dispatchers were instructed to consider any relevant nonclinical factors prior to randomization, and exclude any RCS where these factors would override a clinical determination per the above exclusion criteria. The inclusion criteria were applied automatically (i.e., patients not meeting criteria could not be selected), while exclusion criteria were applied by ambulance dispatchers. Upon selecting the relevant patients and pressing a button in the dispatch interface to compare the selected patients, the RCS was randomly assigned to a study arm via a random number generator. In the control arm, the risk scores for each patient were calculated and stored, but not displayed to the user. In the intervention arm, a marking was displayed in the interface indicating which of the included patients had the highest risk score, along with a color-coded indicator of model confidence (red for high or orange for low, with a cutoff value calibrated based on pilot study data to include ca. 50% of patients in each group). Figure 2 below illustrates the user interface presented to the dispatcher, with Fig 2A illustrating the cases having been selected for inclusion but prior to the user pressing the "predict" button, and Fig 2B illustrating a comparison included in the intervention arm with a high level of model confidence.

In both study arms, the ambulance dispatcher then conferred with the nurses involved in triaging the patients to confirm which patient should receive the available ambulance. In the intervention arm, the ambulance dispatcher noted which patient was proposed by the ML framework, and could access additional information regarding the risk assessment by clicking the risk buttons in the interface. The dispatcher then sent the available ambulance to the patient determined through this process to have the greatest need and cleared the prediction. This process was repeated each time an ambulance became available.

### ML model

The intervention was based on a risk assessment instrument developed by the authors and validated in a previous study [1]. Since the publication of the validation study, the risk assessment instrument was further developed to include free-text notes entered by dispatchers, which were found to improve the performance of the models. The model uses gradient boosting [16] to incorporate structured data regarding patient demographics and findings documented in the CDSS used

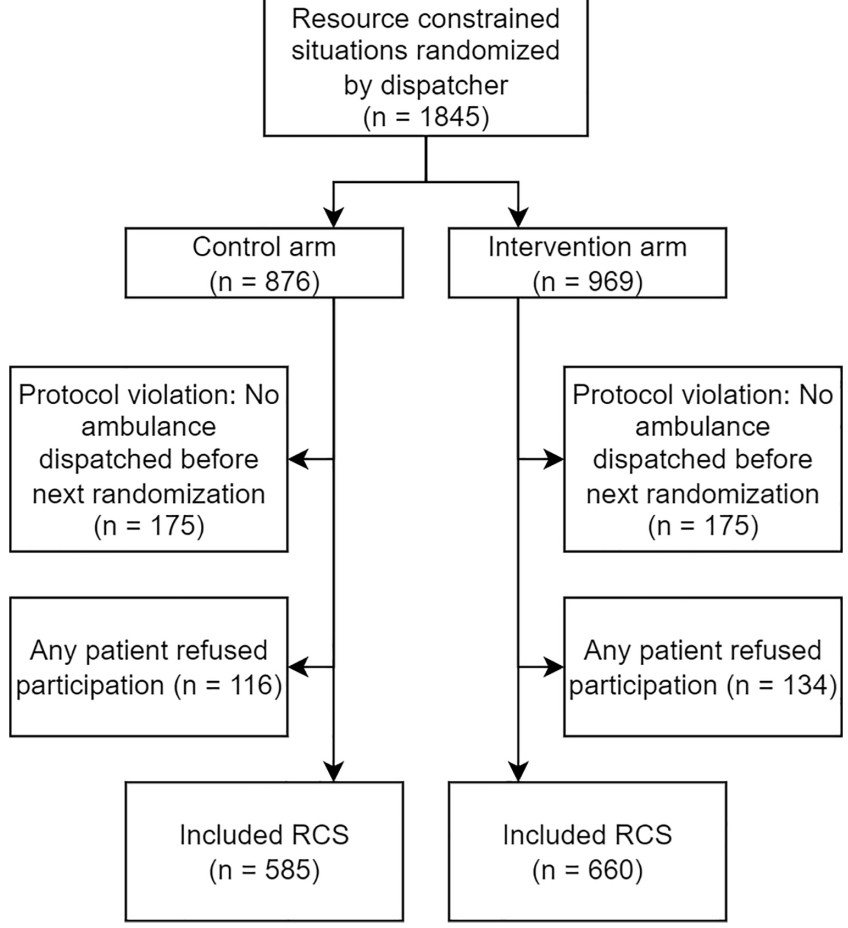

**Fig 1. Participant flow chart.** RCS: Resource Constrained Situation.

at the dispatch centers with unstructured text data coded using a bag-of-words approach to vectorize words and bigrams, resulting in a total of 2,367 predictor variables. Of these, 1,432 encoded words and bigrams with more than 500 occurrences in the free-text, and 861 encoded distinct question/answer combinations recorded in the structured CDSS data, with the remainder encoding the patient call type, age, gender, location, and time of the call. The tree-based gradient boosting architecture was selected owing to its ability to handle substantial interaction effects, and its native capacity to handle the sparse and missing parameters expected to be present in the data. Models were trained on retrospective data collected starting in 2019 and ending in the month prior to implementation of the model in the trial. A full list of included parameters and the gain provided by them in the gradient booting model is included in S1 Supplementary Material – Analysis notebook. The source code of the prediction tool employed in the study is available online under an open-source license [2].

Using this algorithm, the tool separately estimated 4 outcomes thought to be relevant proxy measures of ambulance care need. These were:

1. Abnormal initial primary assessment by the responding ambulance (any value other than normal documented for the patients initial airway, breathing, circulation, and level of consciousness findings).

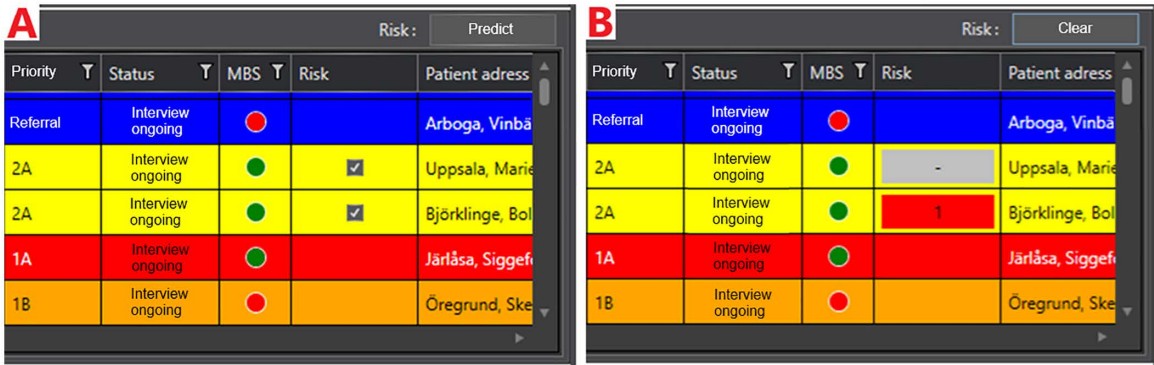

**Fig 2. Illustration of graphical user interface (translated).** Panel **A** illustrates the status of the interface prior to randomization, wherein checkboxes are displayed for patients meeting inclusion criteria including a Priority of 2A or 2B, and a completed decision support form indicated by a green circle in the MBS column. Panel **B** illustrates the status after pressing the "predict" button to perform randomization, here demonstrating an RCS assigned to the intervention group with a high-confidence (red) assessment that the second patient has a greater ambulance need. Additional assessment information could be accessed by clicking the red or gray indicators, and the "Clear" button was then used to reset the interface after dispatching the ambulance. Permission for reproduction granted by system owners.

2. Transport to hospital with lights and sirens.

3. Delivery of prehospital interventions (including catheter placement, medication/oxygen administration, and hospital pre-arrival notification).

4. Admission to hospital in-patient care within 72 hours or mortality within 30 days.

The predicted likelihood for each of the outcomes was then combined into a composite risk score, with the above outcomes weighted at a ratio of 4:2:1:1, respectively, to achieve predictive properties similar to those of NEWS based on an analysis of internal retrospective data. The models were implemented in the openTriage platform and accessed by the dispatching system via an Application Programming Interface [2].

Development of the ML model was frozen upon initiation of the main study phase. There was, however, concern that model performance could degrade over time due to covariate drift stemming from changes in, e.g., documentation practices or the case-mix of patients contacting the dispatch center. Model performance was therefore monitored over the course of the study by the Data Monitoring Committee in terms of the correlation between the patients assigned risk score and their resultant NEWS value without reference to study arm assignment. A risk of degradation was thought to have been identified once early in the study, whereupon the model was retrained with updated data. The model was updated once more during the course of the study upon initiation of the study at the second site. No changes to the models other than using updated data were made. The updates were made on the first of December 2021 and the first of June 2024, respectively.

### Hypotheses

To assess the intervention, a primary and several secondary and ancillary hypotheses were formulated as follows:
Primary:

1. The intervention results in a greater proportion of immediate ambulance responses in RCS being directed to the patient in the most critical condition as operationalized by subsequent NEWS value.

Secondary:

1. The intervention improves differentiation with regards to a composite risk score consisting of ambulance interventions, abnormal initial ambulance findings, emergent transport, hospital admission, and mortality between patients receiving immediate versus delayed ambulance response during RCS.

2. The intervention increases the difference in NEWS between patients receiving immediate versus delayed ambulance response during RCS.

Pre-specified ancillary analyses:

1. Evaluation of overall personnel compliance with risk assessment instrument in intervention arm.

2. Evaluation of compliance in intervention arm cases where the model had a high versus low level of confidence.

3. Evaluation of improved/degraded compliance with risk assessment instrument over time as manifested by a slope change in a time series analysis of intervention group

4. Evaluation of spillover effects as manifested by a significant positive slope in a time series analysis of control group outcomes.

5. Evaluation of change in risk assessment tool predictive value over time (covariate drift).

6. Evaluation of model calibration with regards to age, gender, and complaint category.

Post-hoc ancillary analyses:

1. Evaluation of time to dispatch for patients with an immediately dispatched ambulance.

2. Evaluation of the effect of personnel compliance on the primary hypotheses.

## Outcomes

The primary outcome of the study was the National Early Warning Score (NEWS) of each included patient, based on the first set of vital signs captured by the ambulance crew upon arrival to the patient. Where vital signs were not documented by an ambulance (e.g., if a patient used an alternate mode of transport to the hospital), NEWS component items were multiply imputed including the first set of vital signs documented within 24 hours from the emergency department. NEWS was selected as the primary outcome of the study for two reasons: Firstly, NEWS is widely used in acute care, and has been thoroughly validated as being predictive of outcomes in a variety of adult patient cohorts [17–20]. Secondly, NEWS is based on patient vital signs, and is thus conceptually distinct from, and prior in terms of causality to the outcome measures employed to train the models included in the risk assessment tool.

The first secondary outcome in the study is based on a composite score consisting of each of the four outcomes included in the risk assessment instrument. The second secondary hypothesis consists of an alternate specification regarding the difference in NEWS between prioritized and nonprioritized patients across treatment arms.

## Sample Size

Sample size was determined based on pilot study data indicating that available ambulances were directed to the patient with the highest NEWS 65.3% of the time (considering ties as "correct" assessments), while simulation using randomly selected pairs of potentially eligible patients suggested that the model would mark the patient with the highest NEWS correctly in 70.3% of cases. Using this effect size, an estimated power of 0.8 with an α of 0.05 was achieved at $n \approx 1,500$ using a two-sided test of proportions.

## Randomisation

Patients were recruited by ambulance dispatchers upon identification of an RCS with multiple eligible patients. Study arm allocation at a ratio of 1:1 was performed automatically by the server used to generate risk assessments using a simple random number generator implemented by the numpy python package [21].

## Blinding

Patients and the ambulance/hospital staff collecting outcome data were blind to treatment arm allocation, but by the nature of the intervention dispatchers were aware of the randomization results. Data analysis scripts evaluating the primary and secondary hypotheses were written prior to extracting outcome data using synthetic data simulating the null hypotheses. Where it was necessary to extract outcome data manually (hospital vital sign data at the second study site), the abstractor was blind to treatment group assignment.

## Protocol amendments

No protocol amendments were made during the course of the study. Data collection was terminated prior to collecting the full calculated study sample.

## Statistical analysis

To identify immediate versus delayed ambulance responses, ambulance dispatch logs were analyzed, and the first patient to be assigned an ambulance post-randomization was considered to have been prioritized to receive an immediate response. All other patients involved in the RCS were considered to have not been prioritized, receiving a delayed response. The primary hypothesis was evaluated using logistic regression, with missing vital sign data necessary to calculate NEWS multiply imputed using multivariate imputation by chained equations using the random forest algorithm [22]. Ten sets of imputed NEWS components were generated, and full scores were directly calculated from the component items in line with recommendations [23]. Per the study protocol, outcomes were to be evaluated based on the median value of 5 imputations, but upon further consideration this was felt to risk inflating type-I error rates, and the more rigorous approach of pooling estimates using Rubin's rules was selected by the authors [24]. For the purposes of the analysis, RCS where immediate and delayed responses had the same NEWS were considered as correct assessments. Secondary hypothesis 1 was similarly evaluated using logistic regression, but the second study center had to be excluded due to technical difficulties in gathering comprehensive hospital outcome data. Secondary hypothesis 2 was evaluated using a Wilcoxon rank-sum test applied to the multiply imputed data, pooling $z$-values using Rubin's rules to test for significance [25]. Primary and secondary hypotheses were evaluated based on intention-to-treat.

Six ancillary analyses were pre-specified. Analyses 1–2 investigated compliance with the instrument, and were investigated by assessing the compliance of dispatchers with the risk assessment tool overall and in the high and low confidence groups of the intervention arm, with the hypothesis that compliance would be higher in the high-confidence group. Pre-specified analyses 3–5 regarded changes over time, and were examined using time series analysis within a regression framework employing a variable representing the study month (or months since last model update) as the independent variable of interest. Analysis 6 regarded model calibration, and was conducted by including patient characteristics of interest (age, sex, and major complaint groups) as independent predictors in models evaluating the primary hypothesis. An analysis of loss to follow-up (i.e., patients who withdrew from the study) was performed to examine whether patients who opted out of the study differed from those who remained.

An analysis of time to dispatch was not pre-specified, but was identified as an important indicator of operational efficiency in discussions of the study with clinical and administrative staff. Similarly, an analysis of the effect of non-compliance with the risk assessment tool recommendation was not pre-specified, but considered to be of substantial

importance to understand the impact of dispatcher deviations from model recommendations. An analysis was therefore also performed to investigate hypothetical outcomes if dispatchers had been 100% compliant with the tool to evaluate the potential impact of deviations from model recommendations. Further descriptive post-hoc subgroup analysis of model versus dispatcher prioritization decisions was performed regarding call types. A post-hoc analysis of goodness-of-fit was performed by assessing the models C-indexes in a held-out test dataset from retrospective model training data, and in the trial data [26].

It was anticipated that dispatchers could attempt to circumvent the randomization process (e.g., repeat the randomization if the RCS was assigned to the control arm), and code was implemented to assign repeated randomizations of the same patients to the same treatment arm. In these cases, only the last randomization was included in the analysis. In cases where randomization was repeated but additional patients were included and thus not captured by the repeated randomization check, the last randomization was only included if all randomizations had by chance been assigned to the same treatment arm. All other randomizations were excluded as protocol violations.

All data transformation and analysis were performed using R v4.4.2 [27], and the scripts used to perform all transformations and analysis are available in a public repository [28]. The analysis script used to generate the results presented in this manuscript and its output may also be found as S1 Supplementary Material – Analysis notebook.

## Ethics

Ethical approval for the study was sought and granted by the Swedish Ethical Review Authority (Dnr 2020-00187). An exemption from gathering prospective informed consent from patients was granted for the study by the ethics review board. Informed consent materials were instead mailed to study participants retroactively, at which point patients were given the opportunity to withdraw from the study. The study and its protocol were preregistered at ClinicalTrials.gov (https://clinicaltrials.gov/study/NCT04757194, ID NCT04757194) on 2021-02-17, [29] and are reported according to the CONSORT guidelines [30].

## Results

### Participant flow

Over the course of the study timeframe in each region, a total of 51,782 potentially eligible patients in terms of age, documentation completeness, and priority level were handled by the dispatch centers. Of these, 12,650 had a dispatch time of over 30 minutes, typically indicative of resource constraints in the region. A total of 1845 RCS involving 4,207 patients were randomized by dispatchers. 350 randomizations (19%) were the result of a protocol violation in which no ambulance was dispatched to any patient included in the comparison prior to the next comparison including the same patients. There was also a single case where no ambulance was assigned to any patient in the RCS included in this category of exclusions. Of the remaining 1,495 RCS, 250 (17%) included a patient who upon receiving information about the study, opted to decline participation. Upon applying these exclusions, 1,245 RCS involving 2,764 patients remained for analysis.

### Baseline data

A total of 585 RCS were included in the control arm, and 660 in the intervention arm, corresponding to 1,285 and 1,479 individual patients, respectively. There was thus an imbalance in the randomization both prior to and after applying exclusion criteria. Patient demographics in terms of age and gender were similar across both treatment arms. Overall, dispatch times were similar across both treatment arms, but both dispatch times and times from inclusion to dispatch were shorter for prioritized patients in the intervention group. Note that an analysis of dispatch times was not pre-specified (Table 1).

**Table 1. Patient/dispatch characteristics.**

| | Control | Intervention |
|---|---|---|
| Number of RCS included | 585 | 660 |
| Number of patients included | 1,285 | 1,479 |
| Median age | 77 [76, 78] | 77 [76, 78] |
| Percent female | 45 [43, 48] | 44 [41, 46] |
| Median dispatch time (minutes) | 45 [43, 49] | 45 [42, 47] |
| Median dispatch time for prioritized patient (minutes) | 32 [30, 36] | 29 [27, 31] |
| Median time from inclusion to dispatch for prioritized patient (seconds) | 202 [176, 265] | 96 [82, 135] |
| Average NEWS | 3 [2.8, 3.3] | 2.8 [2.6, 3] |

All values are reported as central tendencies and 95% confidence intervals based on percentiles of 1,000 ordinary bootstrap samples.

## Missingness

The NEWS score calculation was based primarily on ambulance vital parameters, which were missing in between 11.0% (pulse) and 14.2% (temperature) of cases. Missing ambulance data could be due either to patients not receiving an ambulance, or ambulance staff not fully documenting vital parameters. Ambulance vital sign data were supplemented with hospital data collected from the patient's first ED visit within 24 hours of contact with the EMD center, resulting in final missingness rates of between 3.7% (consciousness) and 4.8% (temperature). These combined datasets were used to perform multiple imputation which achieved stable imputed values with good mixing properties across chains. Full data on missingness rates and imputation diagnostics may be found in S1 Supplementary Material.

## Primary and secondary hypotheses

In the control arm, the patient with the highest NEWS value was prioritized in 62.5% of cases, while in the intervention arm the proportion was 68.3%, corresponding to an odds ratio of 1.28 (95% CI [1.00, 1.63], $p = 0.047$). There is thus support for rejecting the null regarding the primary hypotheses of the study. For comparison, random prioritization decisions would have resulted in 55.7% of prioritized patients having the highest NEWS value.

Similarly, intervention effects for the secondary hypotheses regarding composite outcomes achieved statistical significance, with 63.0% of prioritized patients in the control arm having the highest composite outcome score, compared to 69.6% in the intervention arm, corresponding to an odds ratio of OR 1.31 (95% CI [1.01, 1.72], $p = 0.041$) among the 1,106 RCS in the first study site. The average difference between the NEWS value of prioritized and nonprioritized patients in the control arm was 0.62, compared to 1.08 in the intervention arm, corresponding to a mean difference of 0.45, although based on pooled Wilcoxon rank-sum tests, no significant difference could be identified ($p = 0.086$). Results are summarized in Table 2 below.

## Ancillary analyses

Six additional pre-specified analyses were performed to evaluate the properties of the study and the risk prediction model itself.

Per the study protocol, dispatchers were permitted to deviate from the model recommended patient upon conferring with the involved dispatch nurses. In the control arm where information regarding the risk score was not available, the patient with the highest risk score was prioritized in 54.0% (95% CI [50.0, 58.1]) of cases. In the intervention arm, the patient with the highest risk score was prioritized in 80.9% (95% CI [77.7, 83.9]) of cases. To evaluate the impact of these

**Table 2. Summary of primary and secondary hypotheses.**

| Hypothesis | Result |
|---|---|
| Primary: odds of prioritized patient having the highest NEWS in an RCS are greater in the intervention arm per pooled logistic regression | OR 1.28 (95% CI [1.00, 1.63], $p=0.047$) |
| Secondary: odds of prioritized patient having the highest composite outcome score in an RCS are greater in the intervention arm per pooled logistic regression | OR 1.31 (95% CI [1.01, 1.72], $p=0.041$) |
| Secondary: larger difference in NEWS scores between prioritized and nonprioritized patients in intervention arm per a rank-sum test | Z-score 1.716, $p=0.086$ |

deviations, a post-hoc analysis was performed to estimate the intervention effect had the compliance with the risk score been 100% in the intervention group. This analysis identified a substantially stronger effect, with 72.2% of prioritized patients having the highest NEWS value, and resulting in an odds ratio of 1.56 (95% CI [1.21, 2.00], $p<0.001$). A post-hoc subgroup analysis was performed to identify the characteristics of patients preferentially prioritized by the model finding that relative to dispatchers, the model tended to preferentially prioritize patients with call types of difficulty breathing, general elderly, and infection/fever.

An indicator of model confidence was included in the intervention as described in the methods, and while a difference in compliance between the high- and low-confidence groups was identified (81.7% versus 80.3%), the difference was not found to be significant, nor was there a significant difference in outcomes between the groups as assessed per the primary hypothesis. There was however a substantial difference when analyzed in the 100% compliance scenario, with 67.4% of patients assessed by the instrument as highest risk having the highest NEWS value in the low confidence group, versus 78.5% in the high-confidence group, corresponding to an OR of 1.77 (95% CI [1.20, 2.62], $p=0.004$) between the confidence groups. In a post-hoc analysis, the potential effect of blending dispatcher decisions in cases of low model confidence with model decisions in cases of high-confidence was found to be similar to that of the full compliance scenario (71.8% accuracy, OR 1.53 (95% CI [1.188, 1.97])).

A number of analyses investigating temporal effects during the study period were also performed. It was hypothesized that compliance with the tool might change over the study period (either positively due to routinization effects, or negatively due to loss of trust in the tool). No evidence of this was found, with a per-month change in the odds of compliance with the tool in the intervention group being 0.98 (95% CI [0.97, 1.001], $p=0.07$).

It was also hypothesized that the dispatchers might learn from the assessments of the tool, resulting spillover to the control group, manifesting as an increased assessment accuracy over time in the control group. No evidence of this could be found, however, with a per month change in the odds of correct assessment per the primary hypothesis in the control group of 1.00 (95% CI [0.98, 1.01], $p=0.944$).

There was concern that model performance might degrade over time, resulting in a reduced predictive value of the risk assessment algorithm over time. No linear degradation of model performance could, however, be identified, with an average change in Spearman correlation per month since the most recent update of −0.00 (95% CI [−0.01, 0.01], $p=0.906$). In post-hoc analysis comparing model C-indexes in test and trial data, no systematic reduction in model discrimination was identified, with C-indexes ranging from 0.583 to 0.746 across outcomes (see S1 Supplementary Material for a full accounting of model C-indexes).

To evaluate model calibration, indicators for patient age, sex, and clinical category were used as predictors of NEWS together with the ML risk score. We found no residual predictive value of patient age or sex in predicting NEWS when adjusted by the ML risk score. Significant predictive values for patient categories were identified for 2 of the 41 patient

types included in the study (Fever and Difficulty breathing). Full results regarding all ancillary analyses may be found in S1 Supplementary Material.

## Discussion

This study evaluated the ability of a ML-based risk scoring tool to influence care providers at EMD centers, with the aim of improving prioritization decisions among low-priority patients in RCS. The intervention was found to have resulted in an improved differentiation of patients, with 68.3% of prioritized patients in the intervention group having the highest NEWS, versus 62.5% in the control group. The intervention effects were however at the edge of statistical significance, with this difference representing an odds ratio of 1.28 (95% CI [1.00, 1.63]). The intervention appeared to be stable over the nearly 4-year study timeframe, with no signs of degraded model performance. In post-hoc analyses, the intervention group was found to have shorter dispatch delays for prioritized patients (29 versus 32 minutes), suggesting that the intervention increased the speed at which dispatchers made prioritization decisions. A post-hoc analysis also suggested that full compliance with model recommendations in the intervention group would have resulted in a stronger intervention effect with an odds ratio of 1.56 (95% CI [1.21, 2.00]).

In a previous RCT intended for use in identifying cardiac arrests at EMD centers, Blomberg and colleagues [11] identified no intervention effect of an ML-based alerting tool, which was attributed to a lack of compliance with the intervention by EMD nurses. These findings, along with the results of our study identifying the potential for a substantially larger intervention effect had the tool been followed more closely, highlight the need to understand how to build trust in automated risk assessment tools if they are to be used to their full potential. An indicator of model confidence was included in our intervention, but it failed to achieve a statistically significant impact on compliance, despite the high-confidence intervention arm RCS containing a higher proportion of accurate assessments. While the definition of the outcome metrics employed was largely based on clinical judgement, the similar effect sizes with regards to predicting NEWS (OR 1.28) and the composite outcome the model is trained to predict (OR 1.31) suggests that the risk predictions were well aligned with this widely used risk differentiation tool.

The selection of NEWS as the primary outcome measure is a key feature of this study. By selecting an evaluation measure which is not causally dependent on the outcomes used to train the models, the possibility that assignment to the intervention or control arm in and of itself affects the evaluation is minimized. It also addresses issues relating to AI system alignment. As suggested by the orthogonality thesis, the predictive performance of an AI system is thought to be independent of the goals of the system as a whole [31]. Operationalizing the need for a rapid ambulance response in terms of measurable outcomes is difficult, and we cannot assume that the selected proxy measures have done so perfectly. Thus, it is appropriate that both the model and human decisions are evaluated in terms of a measure which in causally independent of either. In this way, the ML framework is not given an unfair advantage over human dispatchers who may have internalized a different definition of patent risk and ambulance care need. While widely used and validated, NEWS may not be perfectly aligned with the patient's prehospital care need—this could explain why the control group accuracy of 62.5% is itself a fairly modest improvement over random prioritization (55.7%). This may however also simply be a reflection of the overall difficulty of triaging potentially emergent conditions over the telephone. That the results of the secondary hypothesis regarding outcomes based on directly measured ambulance and hospital outcomes were similar also suggests that this is the case.

In order to improve the accuracy of the ML models used, we see two general paths: The inclusion of unstructured audio data from the call, and the inclusion of additional structured data from historical patient medical records. The former may be accomplished, e.g., through the processing of audio data from the emergency call using pretrained speech recognition models, and its integration with the decision support system data. The challenges of integrating patient medical record data are primarily legal and technical, necessitating the availability of APIs for obtaining high-quality data from medical records systems in near real-time. Nonetheless, our findings suggest that the effect

size hypothesized based on retrospective validation indeed translated into a real-world impact of similar magnitude, and that applications aimed at higher-risk patient groups using risk assessment tools with similar levels of performance may be safely pursued.

By the nature of the intervention, dispatchers could not be blinded to the treatment assignment. This could explain the uneven distribution between the intervention and control arms both before (969 versus 876) and after (660 versus 585) exclusions were applied, as dispatchers may have sought to repeat the risk scoring if they were assigned to the control group. There is some evidence of this being the case, with 200 of the 350 randomizations having been initially assigned to the control group (chi-squared $p$-value 0.01). While the mechanism behind the unequal randomization is not certain, it does not appear to have resulted in an uneven balance of patient characteristics across the study arms and is thus not likely to bias the results. Missing outcome data also reduced the power of the analysis by inducing between-imputation variability, resulting in an effect size similar to that expected based on pilot study results, but with a substantial degree of uncertainty. The multiple imputation process is also stochastic, and given that the findings are at the very edge of statistical significance, even simply changing the random seed used to generate imputations can variously produce significant or nonsignificant findings. We thus urge that the $p$-values reported here be interpreted thoughtfully as the probability of repeated trials generating an effect at least this extreme under the null hypothesis, rather than dichotomously.

This study was performed at dispatch centers employing nurses in the primary call-taking role. This level of formal education is relatively rare in the context of EMD and could impact the generalizability of the results. It is reasonable to believe that it would be in the direction of underestimating the intervention effects were models of similar precision implemented in a context where the control group was assessed by care providers with less formal training. Similarly, the models used are based on a CDSS used only in a small number of Swedish regions. However, the modeling framework is freely available and can be adapted to structured and free-text data from other CDSS. The study was performed at only two centers, and further studies should evaluate the generalizability of the findings. To limit the risk of patient harm, only patients determined to require a low-priority ambulance response were included in the study. As such, efficacy in higher-acuity patient cohorts, as well as lower priority patients determined not require an ambulance remains to be established.

The study suffered from a degree of post-randomization drop-out due to the 306 RCS (16%) having to be excluded due to at least one patient opting out of the study. The characteristics of patients excluded due to this were however similar to those included in the study (see S1 Supplementary Material), and this source of loss to follow-up thus does not appear to have impacted the findings. It did however reduce the sample size and thus the statistical power of the study.

This randomized controlled trial suggests that ML-based interventions in the context of emergency medical dispatching may have the ability to improve the capacity of care providers to identify patients most in need of an ambulance across a diverse patient cohort. The overall accuracy of the triage process however remains modest, and more can be done to improve the accuracy of the models and the adherence of care providers with model recommendations.

### Patient or public involvement

No formal Patient or Public Involvement efforts were involved in this study.

### Supporting information

**S1 Supplementary Material. Analysis notebook.** 1. Check missingness. 2. Perform multiple imputation. 3. Evaluate multiple imputation. 4. Baseline data. 5. Primary hypothesis. 6. Secondary Hypothesis 1 - Outcome measure. 7. Secondary Hypotheses 2 - NEWS. difference. 8. Evaluate using OLS. 9. Density plot of news differences. 10. Ancillary analyses. 10.1) Evaluation of overall personnel compliance with the risk assessment instrument in intervention arm. a)

Per-model assignment accuracy (post hoc). 10.2) Evaluation of compliance in intervention arm cases where the model had a high versus low level of confidence. a) Confidence groups with per-model assignment (post hoc). 10.3) Evaluation of improved/degraded compliance with risk assessment instrument over time as manifested by a slope change in a time series analysis of intervention group. 10.4) Evaluation of spillover effects as manifested by a significant positive slope in a time series analysis of control group outcomes. 10.5) Evaluation of change in risk assessment tool predictive value over time (covariate drift). 10.6) Evaluation of model calibration with regards to age, gender, and complaint category. 11. Post-randomization loss to follow-up. 12. Descriptive statistics of cases excluded due to patient refusals. 13. Evaluation of study arm imbalance. 14. Simulate accuracy with random assignment. 15. Additional post-hoc analyses. a) Model versus Dispatcher preference analysis. b) Estimate accuracy using model assignments in high-confidence intervention cases. c) Parameter summary/variable importance. d) AUC in retrospective test and trial data.
(HTML)

**S1 Study Protocol. ClinicalTrials Study Protocol.**
(PDF)

**S1 CONSORT Checklist. CONSORT 2025 Statement: updated guideline for reporting randomized trials.** 2025 Hopewell and colleagues. This is an Open Access article distributed under the terms of the Creative Commons Attribution License (https://creativecommons.org/licenses/by/4.0/), which permits unrestricted use, distribution, and reproduction in any medium, provided the original work is properly cited.
(DOCX)

**S1 CONSERVE Checklist. CONSERVE-CONSORT Extension: 2026-02-18.**
(DOCX)

## Acknowledgments

We thank the ambulance dispatchers responsible for executing the study and study participants for their contributions. We also thank Carl Nettelblad at the Uppsala University Department of Information Technology and Ollie Östlund at the Uppsala Centre for Clinical Research for their insight regarding predictive modeling and clinical trial design.

## Author contributions

**Conceptualization:** Douglas Nils Spangler, Hans Blomberg.

**Data curation:** Douglas Nils Spangler, Simon Morelli.

**Formal analysis:** Douglas Nils Spangler.

**Funding acquisition:** Hans Blomberg.

**Investigation:** Douglas Nils Spangler, Hans Blomberg.

**Methodology:** Douglas Nils Spangler, Hans Blomberg.

**Project administration:** Douglas Nils Spangler, Simon Morelli, Lennart Edmark, Hans Blomberg.

**Resources:** David Smekal, Lennart Edmark.

**Software:** Douglas Nils Spangler.

**Supervision:** David Smekal, Hans Blomberg.

**Validation:** Douglas Nils Spangler.

**Visualization:** Douglas Nils Spangler.

**Writing – original draft:** Douglas Nils Spangler.

**Writing – review & editing:** Douglas Nils Spangler, Simon Morelli, David Smekal, Lennart Edmark, Hans Blomberg.

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
