## [Editor Report · Decision Letter 0]

17 Sep 2025

Dear Dr Spangler,

Thank you for submitting your manuscript entitled "Machine learning Assisted Differentiation of Low Acuity patients at Dispatch (MADLAD): A Randomized Controlled Trial" for consideration by PLOS Medicine.

Your manuscript has now been evaluated by the PLOS Medicine editorial staff as well as by an academic editor with relevant expertise and I am writing to let you know that we would like to send your submission out for external peer review.

For clinical studies, please upload a copy of your trial study protocol as a supporting information file. The study protocol should be the version submitted for approval to the institutional review board or ethics committee, should include any amendments to the study protocol, as well as the date of their approval by the institutional review or ethics committee. Please also detail any deviations from the study protocol in the Methods section of your manuscript. The editors will consider the protocol and study conduct prior to a final decision for external review.

Please re-submit your manuscript within two working days, i.e. by Sep 19 2025 11:59PM.

Kind regards,

Jennifer Thorley

PLOS Medicine

---

## [Decision Letter · Decision Letter 1]

19 Dec 2025

Dear Dr Spangler,

Many thanks for submitting your manuscript "Machine learning Assisted Differentiation of Low Acuity patients at Dispatch (MADLAD): A Randomized Controlled Trial" (PMEDICINE-D-25-03177R1) to PLOS Medicine. The paper has been reviewed by subject experts and a statistician; their comments are included below and can also be accessed here: [LINK]

Please accept our sincere apologies for the delay with getting back to you with the decision. As you will see, the reviewers find your study to be of great value for the field but they have raised some concerns that we would ask you to address in full. After discussing the paper with the editorial team and an academic editor with relevant expertise, I'm pleased to invite you to revise the paper in response to the reviewers' comments. We plan to send the revised paper to some or all of the original reviewers, and we cannot provide any guarantees at this stage regarding publication.

Please note that the PLOS Medicine editorial team will be out of office from 20 December to 5 January. During this period, responses to emails and other queries may be delayed.

We ask that you submit your revision by Jan 19 2026 11:59PM. However, if this deadline is not feasible, please contact me by email, and we can discuss a suitable alternative.

Don't hesitate to contact me directly with any questions (kstarowicz@plos.org).

Best regards,

Katarzyna

Katarzyna Starowicz, PhD

Associate Editor

PLOS Medicine

kstarowicz@plos.org

Comments from the reviewers:

Reviewer #1: This was a rare randomised trial of an algorithmic decision aid. Many related prediction models are developed, but few are robustly tested using an RCT, so this research is welcome. The absolute improvement was relatively small, but could translate into meaningful and important benefits if rolled out as standard practice. The results also indicate that greater compliance would lead to larger health benefits.

The trial was pre-registered and had a protocol. It was generally well reported, including the clear labelling of unplanned analysis. The statistical methods used were well justified and appropriate. Being picky, a t-test would probably have been fine for the difference in NEWS values, but it's best to stick with the planned approach.

The trial was relatively complex to conduct given the need to only randomise mid-range risk patients because of ethical concerns. This was well explained by the inclusion and exclusion criteria. I'm glad the authors chose individual randomisation over cluster randomisation.

There was an imbalance in randomisation, which is explained in the discussion (first paragraph of limitations section) but should probably come in the results section. Also, the CONSORT flow chart implies that the imbalance happened at the randomisation stage, which made me think that the study did not use a 1:1 ratio. However, some randomisations were excluded, so this should be included in the flow chart.

Minor comments

- The Figure 1 legend could be more detailed. What does MBS mean?

- How many variables are used by the machine learning tool (page 7, line 17)? It would be good to get an idea of the data needs of the tool as an indicator of its relatively complexity. This would help understand problems like the potential spillover effect (page 12).

- ML can also mean "maximum likelihood", so it may be best to spell it out. I also had to check RCS a few times, so I'd spell that out too.

- The 1:1 randomisation ratio should probably be mentioned in the methods.

- I did find the need to retrospectively consent patients as strange and it impacted the power of the study. I can see the argument for empowering patients, but without being able to robustly test such systems, improvements for all patients will be slower. Potentially this was beyond the control of the investigators.

- Is there any qualitative data from the ambulance staff concerning their attitudes towards the tool and the trial?

- Line 29, page 4 "with a cutoff value calibrated to include ca. 50% of patients in each group". I presume this means the randomised groups?

- Table 1 column says "percent female" but the statistic given is a proportion. The second statistics could be presented as integers as a fraction of a second feels like spurious accuracy.

- Page 12 "There was also concern that model performance could degrade over time", perhaps add a reason why this could happen.

Reviewer #2: A randomised controlled trial of the ability of a machine learning versus a heuristic algorithm to determine ambulance deployment to the higher acuity patient (as determined by first NEWS score) for lower risk patients, when two or more patients required an ambulance response.

The authors are to be commended on completing a taxing study and providing a thorough explanation both of the study and its findings. The study is both innovative and interesting.

Significant issues

A detailed description of the ML algorithm being used needs to be included in the paper/supplementary material along with the changes made during the study. Statistical analysis lines 17 to 20 P7 should form part of this but are insufficient.

The primary outcome is of borderline significance and should be reported as such, as indeed they have done in the discussion. This is particularly important given the missing data from the outcome and resultant imputation. I think this would be clearer that this was the understanding of the authors in the abstract if they simply added "may" to the abstract discussion "This study suggests that clinical ML-based decision support tools may have the ability"

The paper would benefit from a definitions section.

The definition of a resource constrained situation in the paper is probably best found in the inclusion criteria - I believe from the results and inclusion criteria that on some occasions there are more than two patients requiring one ambulance - but this would be better in a definitions section

The authors do not appear to have defined the difference between immediate and delayed ambulance response or explained how this would have been affected by the intervention this makes these results difficult to interpret

Minor issues

P2 lines 10-12 risk the authors having missed relevant studies and could be re-worded to remove this risk.

Fig 1 needs more careful explanation as part of the figure caption - I think it means that the second yellow patient has been marked as high confidence higher acuity in comparison to the first yellow patient - but don't know what the mbs column means and I took a while to realise the risk column referred to the algorithm. What dose Risk:clear and risk:predict refer to and where are the risk buttons which allow extra information to be ascertained?

P6 lines 12-28 is a discussion point more than methods

The information on model updating in results should come near the start - before the main results - so that the reader is aware of what is being assessed.

Reviewer #3: Thank you for submitting this paper for review. The paper is well thought out and well written, and targets a specific use case that is relevant and not previously explored. I applaud the authors for making a lot of effort to be thoughtful about what metrics and outcomes to use, as well as detailing and addressing potential concerns related to the model in their ancillary analysis section.

I agree with the authors' conclusions based on the results that they have shared. The only comment I have is that it might be interesting to learn more about why dispatchers made different decisions from the model itself. What caught my eye was that the difference between the control and intervention arms could have been much larger had the dispatcher followed the model's recommendation each time, so I'd be curious if there's any insight to be shared on why the dispatchers sometimes overruled the model incorrectly (at least based on NEWs). If they did this inappropriately, without obvious benefit to the patient, this could be an interesting finding to highlight more. For example, it might also be worth discussing if it makes more sense to use the model recommendations only (in some/most/all) situations, rather than have a dispatcher be able to override it, if that will ultimately lead to better resource allocation.

Reviewer #4: Thank you for the opportunity to review PMEDICINE-D-25-03177R1: Machine learning Assisted Differentiation of Low Acuity patients at Dispatch (MADLAD): A Randomized Controlled Trial.

I found this to be a well-designed study and well-written paper that was engaging and enjoyable to read, with the authors generally doing a great job at explaining their logic.

The conclusions are nicely measured (i.e. nothing understated or overstated) in relation to their findings. Overall, this is one of the better papers I've reviewed for some time, and I feel it will make a valuable contribution to the literature.

That said, I do have the following points that I think need addressing before this paper should be considered suitable for publication.

Comment 1. On reading the introduction section a couple of times, I feel that the description and identify of the pre-existing Machine Language model that forms the basis of the paper, is very unclear for the reader. As best I can tell, this model is referred to on Page 2, Lines 8-10, where the authors say "In previous research, an open source ML-based risk assessment tool was developed and retrospectively validated in a cohort of patients receiving ambulance care in the region of Uppsala." However it's unclear to the reader whether the authors themselves developed this model, or whether they are simply referring to a model in the literature, developed by others. In addition, it's unclear at this point in the paper that this very same model is the one that the authors use as the basis for the submitted paper. This only becomes clear to the reader much later in the paper, on Page 5, Line 12. This issue needs to be clearer for the reader much earlier in the paper - within the introduction, but them re-emphasized early in the methods section.

Comment 2. Building on from Comment 1 above, the paper needs a brief description of the machine learning algorithm that the authors previously developed and use as the basis for this paper, so that the current paper is more self-contained, in not requiring the reader to consult their previous paper to get a basic understanding on the algorithm. In particular, it would be great to include a brief description of the data that are used as inputs (i.e. what dispatch information). On Page 5, Line 13, the authors state that the "the risk assessment instrument was further developed to include free-text notes entered by dispatchers", however I didn't see a description of the other predictors. My apologies if I've somehow overlooked these being mentioned.

Comment 3. In relation to some Resource Constrained Situations (RCSs) having more than 2 emergency calls to triage, it would be informative to know what the null expectation is, of the first assigned call having the highest NEWS score if this process was random. While this null expectation would be exactly 50% when there are only 2 calls per RCS, this percentage will be lower when there are sometimes more calls per RCS. Knowing these expected percentages would be valuable in the paper, in helping the reader to understand how well the intervention and standard method are working in relation to randomization.

Comment 4. Regarding the description of participant inclusion and exclusion criteria (Page 4), it would be better to have this with a brief introductory sentence (at the moment this drops the reader abruptly into the heading of "Inclusion Criteria", followed by point form). Otherwise, given that the hypotheses at the end of the introduction are also presented in point form, there is a risk that the reader gets the unfortunate impression that the authors have lost interest in structured paragraphs, which, in turn, risks losing the reader's interest. Note: I also think that study location and study period should be included in the description of inclusion/exclusion criteria.

Comment 5.

One of the really important and interesting results in this study was that predictive accuracy would have been higher if dispatchers had followed the predictions from the machine-learning algorithm. I.e. that the net effect of manual overrides was to reduce predictive accuracy. The authors do a nice job of discussing this in Page 13, Lines 23-28. I also appreciate the diplomacy required in this commentary, and that the authors have worded this in a really constructive way. However I do think that from an analytical / study design point of view, this particular topic needs a bit more. Firstly, this component of the analysis, despite its importance, was not explicitly listed as one of the hypotheses or ancillary analyses. While the authors mention it as on Page 8, Lines 6-8 as ("An analysis was also performed to investigate the hypothetical outcomes if dispatchers had been 100% compliant with the tool to evaluate the potential impact of compliance rates."), it has diminished prominence for the reader given that it's not listed one of the numbered analyses. In addition, the issue raised by this important additional analysis remains unresolved for the reader, in that ultimately the important question is whether the manual overriding by ambulance directors significantly impacted on triage accuracy. I.e. this should be explicitly tested. While I appreciate that these may now represent post-hoc analyses, it's still important to list them with the same prominence as other hypotheses/analyses, albeit declared as post-hoc. In relation to this, this particular analysis (of testing if the ML-only accuracy differed from the overridden accuracy) is something that should definitely be 2-tailed, as it is conceivable that manual overrides could have in fact improved the triage accuracy.

Finally, as a suggestion for the authors (not necessarily something that needs to be included in the paper), an interesting post-hoc analysis could be to examine the final accuracy under the scenario that overrides are only permitted in circumstances where the model confidence was low. This could provide something of a "best of both worlds" approach, of using expert opinion in situations where the model can't differentiate. The authors might like to consider whether such ideas could be mentioned in the discussion among the suggestions for potential model improvement.

Comment 6. I think the Discussion section needs some brief mentions of the key statistics early in the discussion, so that the Discussion is more stand-alone in conveying the numeric magnitude of the main findings.

Comment 7. Page 7, Line 26 has an erroneous line break.

Comment 8. Page 9, Line 14 refers to Figure 1, but should refer to Figure 2.

---

* Please upload any figures associated with your paper as individual TIF or EPS files with 300dpi resolution at resubmission; please read our figure guidelines for more information on our requirements: http://journals.plos.org/plosmedicine/s/figures. While revising your submission, we strongly recommend that you use PLOS's NAAS tool (https://ngplosjournals.pagemajik.ai/artanalysis) to test your figure files. NAAS can convert your figure files to the TIFF file type and meet basic requirements (such as print size, resolution), or provide you with a report on issues that do not meet our requirements and that NAAS cannot fix.

After uploading your figures to PLOS's NAAS tool - https://ngplosjournals.pagemajik.ai/artanalysis, NAAS will process the files provided and display the results in the "Uploaded Files" section of the page as the processing is complete.

If the uploaded figures meet our requirements (or NAAS is able to fix the files to meet our requirements), the figure will be marked as "fixed" above. If NAAS is unable to fix the files, a red "failed" label will appear above.

When NAAS has confirmed that the figure files meet our requirements, please download the file via the download option, and include these NAAS processed figure files when submitting your revised manuscript.

* Please ensure that the study is reported according to the CONSORT 2025 guideline and include the completed CONSORT 2025 checklist as Supporting Information. When completing the checklist, please use section and paragraph numbers, rather than page numbers. Please add the following statement, or similar, to the Methods: "This study is reported as per CONSORT guideline (S1 Checklist)."

FIGURES AND TABLES

SUPPLEMENTARY MATERIAL

REFERENCES

RCTs

* PLOS Medicine requires that all trials be prospectively registered in one of registries recognized by WHO. Please explain in the methods section why the clinical trial registration date is retrospective (2 days after recruitment of the first participant). Please ensure that these details are included in the Methods section.

* Please structure the Methods section using the following sub-headings: Study design and participants, Randomization and masking, Procedures, Outcomes, Statistical analysis.

* If the outcomes were not prespecified in the protocol, please define them in the Methods (Outcomes section) as post hoc and explain why they were added. Post-hoc comparisons should be presented as hypothesis generating rather than conclusive.

* Please ensure that all prespecified outcomes (primary, secondary, and exploratory) are listed in the Methods/Outcomes section and indicate whether there are outcomes that are not presented in the current report.

* Please specify the dates (Month Day, Year) during which study enrollment and follow up occurred.

* Please include absolute numbers wherever you report percentages; eg, n/N (%)

* Please present the safety data for the study including numbers of specific events and whether or not adverse events are thought to be related to treatment. AEs should be reported in the abstract, per CONSORT and CONSORT-Harms.

* Please complete the CONSORT 2025 checklist (https://www.equator-network.org/reporting-guidelines/consort/) and ensure that all components of CONSORT are present in the manuscript, including how randomization was performed, allocation concealment, blinding of intervention, definition of lost to follow-up, power statement. When completing the checklist, please use section and paragraph numbers, rather than page numbers.

* Please report your abstract according to CONSORT for abstracts, following the PLOS Medicine abstract structure (Background, Methods and Findings, Conclusions) https://www.equator-network.org/reporting-guidelines/consort-abstracts/

* If your trial had to undergo important modifications in response to extenuating circumstances, please complete the CONSERVE-CONSORT checklist and provide in your Supporting Information; (https://www.equator-network.org/reporting-guidelines/guidelines-for-reporting-trial-protocols-and-completed-trials-modified-due-to-the-covid-19-pandemic-and-other-extenuating-circumstances-the-conserve-2021-statement/). When completing the checklist, please use section and paragraph numbers, rather than page numbers.

* In keeping with our commitment to Open Science, please include the study protocol document and analysis plan (including any amendments) as Supporting Information to be published with the manuscript if accepted.

* Please note that PLOS Medicine requires prospective, public registration of a data sharing plan (as part of mandatory clinical trials registration) for all clinical trials that began enrollment on or after January 1, 2019, in accordance with ICMJE requirements.

---

## [Decision Letter · Decision Letter 2]

11 Feb 2026

Dear Dr. Spangler,

Thank you very much for re-submitting your manuscript "Machine learning Assisted Differentiation of Low Acuity patients at Dispatch (MADLAD): A Randomized Controlled Trial" (PMEDICINE-D-25-03177R2) for review by PLOS Medicine.

I have discussed the paper with my colleagues and the academic editor and it was also seen again by 3 reviewers. I am pleased to say that provided the remaining editorial and production issues are dealt with we are planning to accept the paper for publication in the journal.

[LINK]

We look forward to receiving the revised manuscript by Feb 18 2026 11:59PM.

Sincerely,

Katarzyna Starowicz, PhD

Associate Editor

PLOS Medicine

kstarowicz@plos.org

Requests from Editors:

GENERAL EDITORIAL REQUESTS

* At this stage, we ask that you include a short, non-technical Author Summary of your research to make findings accessible to a wide audience that includes both scientists and non-scientists. The Author Summary should immediately follow the Abstract in your revised manuscript. This text is subject to editorial change and should be distinct from the scientific abstract. Ideally each sub-heading should contain 2-3 single sentence, concise bullet points containing the most salient points from your study. In the final bullet point of ‘What Do These Findings Mean?’ Please include the main limitations of the study in non-technical language.

Please see our author guidelines for more information: https://journals.plos.org/plosmedicine/s/revising-your-manuscript#loc-author-summary."

* Please confirm that your title complies with PLOS Medicine's style. Your title must be nondeclarative and not a question. It should begin with main concept if possible. "Effect of" should be used only if causality can be inferred, i.e., for an RCT. Please place the study design ("A randomized controlled trial," "A retrospective study," "A modelling study," etc.) in the subtitle (ie, after a colon).

* Please confirm that your abstract complies with our requirements, including format (three sections: Background, Methods and Findings, and Conclusions) and providing all the information relevant to this study type https://journals.plos.org/plosmedicine/s/submission-guidelines#loc-abstract

* Please ensure that the Introduction ends with a clear description of the study question or hypothesis.

* Please ensure that all abbreviations are defined at first use throughout the text.

* Please confirm that all numbers presented in the abstract are present and identical to numbers presented in the main manuscript text.

* Please review your text for claims of novelty or primacy (e.g. 'for the first time') and remove this language. In addition, please check that any use of statistical terms (such as trend or significant) are supported by the data, and if not please remove them.

* Please remove the 'conclusions' subheading from the discussion. Please also remove any other subheadings from the discussion.

* Statistical reporting: Please revise throughout the manuscript, including tables and figures.

- Please report statistical information as follows to improve clarity for the reader ""22% (95% CI [13,28]; p</=)"".

- Please separate upper and lower bounds with commas instead of hyphens as the latter can be confused with reporting of negative values.

- Please repeat statistical definitions (HR, CI etc.) for each set of parentheses."

* In the abstract, please include the important dependent variables that are adjusted for in the analyses.

* In the author summary, please revise formatting and ensure you use bullet points.

* In the author summary, in the final bullet point of 'What Do These Findings Mean?', please include the main limitations of the study in non-technical language.

* The funding statement should include: specific grant numbers, initials of authors who received each award, URLs to sponsors’ websites. Also, please state whether any sponsors or funders (other than the named authors) played any role in study design, data collection and analysis, the decision to publish, or preparation of the manuscript. If they had no role in the research, include this sentence: “The funders had no role in study design, data collection and analysis, decision to publish, or preparation of the manuscript.”

* All authors must declare their relevant competing interests per the PLOS policy, which can be seen here: https://journals.plos.org/plosmedicine/s/competing-interests For authors with ties to industry, please indicate whether any of the interests has a financial stake in the results of the current study.

* PLOS defines the “minimal data set” to consist of the data set used to reach the conclusions drawn in the manuscript with related metadata and methods, and any additional data required to replicate the reported study findings in their entirety. Authors do not need to submit their entire data set, or the raw data collected during an investigation. Please submit the following data:

The values behind the means, standard deviations and other measures reported;

The values used to build graphs;

The points extracted from images for analysis."

* For studies in which a novel model is central to the manuscript's findings, as is the case here, authors are responsible for providing the source code needed to replicate the study's findings in a repository (such as GitHub, SourceForge or Bitbucket) or a cloud computing service (such as Code Ocean). Protection of authors’ intellectual property will not be cause for exception. Please explain in the manuscript’s Data Availability Statement how readers can access the shared code.

* Thank you for agreeing to make your data available. At this time, please provide the link to the data repository and accession numbers required for access.

* Please include the statement on code availability in the data availability statement.

* Figures cannot be reproduced from other sources that are not CC-BY. Please provide a different figure.

* Please provide titles and legends for all figures and tables (including those in Supporting Information files). Please define all acronyms used in each figure or table in its corresponding legend.

* Please ensure that where relevant figures include 95% CIs.

* When a p value is given, please specify the statistical test used to determine it in the legend.

* Please consider avoiding the use of red and green in order to make your figure more accessible

* Please consider if moving some of the figures shared in the supplementary information into the main text would aid the reader.

* Please include an Acknowledgments section in your manuscript.

* Please include an acknowledgment of study participants.

* Please include in the Acknowledgments recognition of individuals who played a role in data collection or participant care or involvement

* Please complete the CONSORT 2025 checklist and ensure that all components of CONSORT 2025 are present in the manuscript, including how randomization was performed, allocation concealment, blinding of intervention, definition of lost to follow-up, power statement. When completing the checklist, please use section and paragraph numbers, rather than page numbers. The checklist should be included as supporting information, and should be cited in the article.

* As your trial had to undergo important modifications in response to extenuating circumstances, please complete the CONSERVE-CONSORT checklist and provide in your Supporting Information.

When completing the checklist, please use section and paragraph numbers, rather than page numbers."

* PLOS Medicine requires that all trials be prospectively registered in one of registries recognized by WHO. Please provide information on study registration in the Methods section.

* Some of the outcome measures or methods appear to differ between the submitted manuscript and the trial registry and/or protocol. Please clarify and explain the discrepancy. If the outcomes were not prespecified in the protocol, please indicate that they were post hoc and explain why they were added. Post hoc comparisons should be presented as hypothesis generating rather than conclusive.

* In accordance with ICMJE requirements, PLOS Medicine requires prospective, public registration of a data sharing plan (as part of mandatory clinical trials registration) for all clinical trials that began enrollment on or after January 1, 2019.

* The sample size listed in the submitted manuscript and the trial registry differ. Please explain the discrepancy.

* The main analysis should be intention to treat (ie, all individuals randomized are included in the analysis in the groups to which they were originally assigned. If the study included dropouts, specify whether their data are imputed and if so using what method. Please refer to as modified ITT).

* The CONSORT flowchart should be figure 1, please revise.

* Please present the safety data for the study including numbers of specific events and whether or not adverse events are thought to be related to treatment.

* Please report your abstract according to CONSORT for abstracts, following the PLOS Medicine abstract structure (Background, Methods and Findings, Conclusions) https://www.equator-network.org/reporting-guidelines/consort-abstracts/

Per CONSORT, please note that only the primary outcome of the trial should be reported in your Abstract. Secondary outcomes should only be included in the Abstract if all secondary outcomes are fully reported. For trials that have many secondary outcomes, the Abstract should be limited to reporting the primary outcome.

For more details on the requests below please see Geoffrey P Garnett, Simon Cousens, Timothy B Hallett, Richard Steketee, Neff Walker. Mathematical models in the evaluation of health programmes. (2011) Lancet DOI:10.1016/S0140-6736(10)61505-X.

* Please provide a complete list of model parameters, including clear and precise descriptions of the meaning of each parameter, together with the values or ranges for each, with justification or the primary source cited, and important caveats about the use of these values noted.

* Please provide a clear statement about how the model was fitted to the data including where relevant goodness-of-fit measure, the numerical algorithm used, which parameter varied, constraints imposed on parameter values, and starting conditions.

* For uncertainty analyses, please state the sources of uncertainties quantified and not quantified this can include parameter, data, and model structure.

* Please provide sensitivity analyses to identify which parameter values are most important in the model. Uncertainty estimates seek to derive a range of credible results on the basis of an exploration of the range of reasonable parameter values. The choice of method should be presented and justified.

* Please discuss the scientific rationale for this choice of model structure and identify points where this choice could influence conclusions drawn. Please also describe the strength of the scientific basis underlying the key model assumptions.

Comments from Reviewers:

Reviewer #1: This is an interesting and well-conducted trial. I had only minor comments in my first review and the authors have answered them well.

One minor comment that the is more for interest than addressing. Line 19, page 16. "and this source of loss to follow-up thus does not appear to have impacted the findings", this loss likely did not bias the findings, but it did increase uncertainty. It would be interesting to show the ethics committee the impact of this loss. Whilst I think the trial shows good evidence of the intervention being useful in practice, others may want more certainty (often meaning a smaller p-value). The wider costs of this missing data could then become large if the decision to implement is delayed.

Reviewer #2: The authors have done a great job of answering my comments - and as I noted before have done a really tough study well.

They now have good definitions of "resource-constrained" and immediate vs delayed - but these are in the introduction and at the start of the stats section. I still think these would be better placed in the methods section together under a definitions subheading perhaps just before "statistical analysis" - but this is a very minor point.

Reviewer #4: I appreciate the authors' thoughtful response to my comments and those of the other reviewers. I am satisified with the revised version. Well done to the authors on a great paper that I think will be well received by readers.

[LINK]

---

## [Editor Report · Decision Letter 3]

3 Mar 2026

Dear Dr. Spangler,

Thank you very much for re-submitting your manuscript "Machine learning Assisted Differentiation of Low Acuity patients at Dispatch (MADLAD): A Randomized Controlled Trial" (PMEDICINE-D-25-03177R3) for review by PLOS Medicine.

I have discussed the paper with my colleagues from editorial team and there are few minor editorial comments you need to address before we can proceed with acceptance of your manuscript.

[LINK]

We expect to receive your revised manuscript within 3 days. Please email us (kstarowicz@plos.org or plosmedicine@plos.org) if you have any questions or concerns.

We look forward to receiving the revised manuscript by Mar 06 2026 11:59PM.

Sincerely,

Katarzyna Starowicz, PhD

Associate Editor

PLOS Medicine

plosmedicine.org

Requests from Editors:

- Please remove the MADLAD abbreviation in the title, given that it's not used (i.e. an abbreviation should be included in an Abstract, for example, if it's used again in the Abstract. But if not, the abbreviation is unnecessary. So the title could be : Machine learning Assisted Differentiation of Low Acuity patients at Dispatch: A Randomized Controlled Trial"

or

Machine learning Assisted Differentiation of Low Acuity patients at Dispatch: The MADLAD Randomized Controlled Trial"

- line 12: parallel-group is usually used, not 'parallel-grouped'

- The abstract format is wrong - you separated methods and findings section, clinical trial nr is in methods sections not in last line of conclusion, and no limitation of study in last line of M&F (for guidelines please have a look at https://journals.plos.org/plosmedicine/s/submission-guidelines#loc-abstract)

- You need to state in the Abstract that the study was performed at 2 sites. This should also be noted as a limitation.

- You need to identify the primary outcome in the Abstract, and secondary outcomes (the latter if all are reported). This is essential for RCTs.

- Please also report the results for these primary and secondary outcomes/objectives.

- line 20: should NEWS be NEWS 2, as per line 17?

- line 12-13--sentence is not grammatically correct. Suggest they add, "We performed" before "a parallel-group, randomized...."

- Numbering needs to be contiguous--i.e. you should not restart numbering on each page.

- last bullet point of the author summary:" A smaller than expected sample size entailed that results were at the edge of statistical

22 significance, limiting the certainty of the findings." Ordinarily I would suggest to revise to something along the lines of "Due to the small sample size, these findings should be validated in a larger cohort."

- Note that the authors say in the methods that they did not meet their sample size--this needs to be acknowledged in the Abstract as it's a major study limitation.

- Pertaining to the above, you need to explain a bit more why you didn't recruit the 1500 RCS that they had intended. Please expand 'administrative reasons' in the manuscript (not only as included in rebuttal).

- We would not normally have the aim and hypotheses section as part of the Introduction in the way you have laid it out. The Intro should end with a sentence explaining the study question. The primary and secondary objectives/outcomes should be ID'd in the Abstract. You can include the additional study detail in the Methods section. not as written in the Intro.

- Line 7, page 13 " between 3.7% (consciousness) and 4.8% (temperature)." I am not understanding what these percentages mean.

- do you have a table with criteria assessed by a dispatcher and criteria incorporated into the assessment by the AI tool? I think this would be useful to see.

I put deadline for the end of the week. If you need more time please let me know.

[LINK]

---

## [Editor Report · Decision Letter 4]

9 Mar 2026

Dear Dr Spangler,

On behalf of my colleagues and the Academic Editor, Jean-Louis Vincent, I am pleased to inform you that we have agreed to publish your manuscript "Machine learning Assisted Differentiation of Low Acuity patients at Dispatch: The MADLAD Randomized Controlled Trial" (PMEDICINE-D-25-03177R4) in PLOS Medicine.

PRESS

Sincerely,

Katarzyna Starowicz, PhD

Associate Editor

PLOS Medicine